# Differences in COVID-19 Risk by Race and County-Level Social Determinants of Health among Veterans

**DOI:** 10.3390/ijerph182413140

**Published:** 2021-12-13

**Authors:** Hoda S. Abdel Magid, Jacqueline M. Ferguson, Raymond Van Cleve, Amanda L. Purnell, Thomas F. Osborne

**Affiliations:** 1VA Palo Alto Healthcare System, US Department of Veterans Affairs, Palo Alto, CA 94304, USA; Jacqueline.Ferguson@va.gov (J.M.F.); Raymond.VanCleve@va.gov (R.V.C.); Thomas.Osborne@va.gov (T.F.O.); 2Department of Epidemiology and Population Health, Stanford University, Stanford, CA 94305, USA; 3Stanford Center for Population Health Sciences, Stanford University School of Medicine, Stanford, CA 94305, USA; 4Center for Innovation to Implementation, Veterans Affairs Palo Alto Health Care System, MDP-152, 705 Willow Road, Menlo Park, CA 94025, USA; 5Veterans Affairs Central Office, Washington, DC 20420, USA; Amanda.Purnell@va.gov; 6Department of Radiology, Stanford University School of Medicine, Stanford, CA 94305, USA

**Keywords:** Veterans, COVID-19, social determinants of health, county-level, race, health disparities

## Abstract

COVID-19 disparities by area-level social determinants of health (SDH) have been a significant public health concern and may also be impacting U.S. Veterans. This retrospective analysis was designed to inform optimal care and prevention strategies at the U.S. Department of Veterans Affairs (VA) and utilized COVID-19 data from the VAs EHR and geographically linked county-level data from 18 area-based socioeconomic measures. The risk of testing positive with Veterans’ county-level SDHs, adjusting for demographics, comorbidities, and facility characteristics, was calculated using generalized linear models. We found an exposure–response relationship whereby individual COVID-19 infection risk increased with each increasing quartile of adverse county-level SDH, such as the percentage of residents in a county without a college degree, eligible for Medicaid, and living in crowded housing.

## 1. Introduction

Disparities in COVID-19 infection and mortality vary across the U.S. [1,2,3] These disparities, particularly among racial and ethnic minorities, may be driven by factors such as area-level social determinants of health (SDH) and structural resources [4,5,6,7,8,9,10,11,12,13,14,15,16]. For example, higher income inequality at state [17] and county [18] levels has been linked to increased COVID-19 burden. Income inequality may exacerbate infection risk, as the most disadvantaged individuals have different working environment opportunities compared to other resident in the same community [19]. More specifically, lower-income individuals are more likely to have public-facing jobs such as service, child and elder care, and cleaning/janitorial services, and these individuals are also are more likely to reside in crowded housing, further increasing the risk of exposure [20]. Although these studies have provided evidence necessary to understand the area-level SDH associated with COVID-19 disparities, the examination of these relationships in integrated healthcare systems has been limited. To our knowledge, the association between county-level SDH and COVID-19 among Veterans has not been previously examined.

Health systems are a focal point of the COVID-19 pandemic. They are vital to understanding the extent of the pandemic and identifying groups at highest risk. The electronic health record database of the Department of Veterans Affairs (VA) offers the single largest national data resource available with the necessary information on system-wide testing and detailed medical histories to examine racial and ethnic disparities in the U.S. Recent research suggests that some racial and ethnic minorities, as well as socioeconomically disadvantaged groups within the VA, are bearing a disproportionate burden of COVID-19 [21,22,23].

Identifying the area-level social determinants of health affecting this COVID-19 burden is critical to implementing an effective healthcare system response strategy. Detailed geographic data on social inequities in COVID-19 outcomes stratified by demographics are required, and understanding the distribution of area-specific characteristics is critical to mounting an adequate, timely, and comprehensive response in the pandemic, in addition to providing an evidence base for policy and resource allocation. Leveraging the rich electronic health records of the VA, in this report, we combine electronic health record (EHR) data from the VA with county-level characteristics to assess associations between area-level SDH and COVID-19 infection risk among Veterans with the goal of optimizing care and prevention strategies for our patients.

## 2. Materials and Methods

We retrospectively examined records from Veterans actively enrolled in VA who were tested for SARS-CoV-2 at VA between 8 February 2020 and 28 December 2020. Methods have been previously described in detail [23]. In brief, we included demographic characteristics from the VA’s EHR database and utilized the Veteran’s home zip code to geographically link publicly available area-based SDH, as it has previously been identified as critical for COVID-19 health equity in previous literature [1,20,24,25,26,27]. A detailed table describing each county-level SDH, source, and original variable name from the source is provided in Appendix A. We categorized each area-based SDH into quartiles according to the positive case distribution in our analytic sample. Correlations between each county-level SDH are presented in Appendix A. We excluded Veterans missing county-level SDH, as well as one VHA facility with fewer than 5 COVID-19 positive cases. The final analytic sample comprised 778,599 Veterans.

### Statistical Analysis

We utilized generalized linear models to report risk ratios and 95% confidence intervals for the risk of testing COVID-19 positive in relationship to SDH. To examine effect modification by race between SDH and COVID-19 positivity risk, we stratified our analysis by race, including White, Black, and Other Veterans (includes Asian, American Indian/Alaska Native, Native Hawaiian/Other Pacific Islander). All models were adjusted for individual demographics, facility characteristics, state, and other SDH characteristics that are important for health equity but not identified a priori as primary SDH characteristics of interest [1,20,26,27,28,29]. Model standard errors are clustered by VA facility. We conducted all statistical analyses using Stata Version 15 (StataCorp LLC, College Station, TX, USA). This quality assessment project received a Determination of Non-Research from Stanford Institutional Review Board as well as by VA determination.

## 3. Results

As of 28 December 2020, among the 779,599 Veterans tested at VA, 77,692 (10%) tested positive for COVID-19 (Table 1). On average, compared with White Veterans, Black and Other Veterans lived in counties with higher percentages of non-US born residents; with a higher percentage of non-White residents; individuals without a high school diploma; persons receiving food stamps/SNAP benefits; persons living in crowded housing; persons without broadband; persons living in multigenerational housing (i.e., households where grandparents have children who are under 18); and persons in deep poverty. In addition to examining county-level contextual factors, fully adjusted models adjusted for individual demographics and facility characteristics, Gini coefficient, percentage aged 65+ living alone, rural/urban/highly rural, unemployment rate (2017), and without health insurance (data not shown). Consistent with our previous findings, in fully adjusted models we found that female, Black, urban, low-income, and disabled Veterans were more likely to test positive for COVID-19 [23]. These disparities are also consistent with other studies examining differences in COVID-19 testing and test positivity within the VA population [2,21].

We found an exposure–response relationship with individual infection risk of COVID-19 increasing with each increasing quartile of adverse county-level SDH for the following SDH variables; percentage of residents in a county without a college degree, percentage eligible for Medicaid, and the percentage of residents living in crowded housing (Table 2). The risk of testing positive for COVID-19 among Veterans living in counties, with the top quartile of percentage of residents without a college degree compared to Veterans living in counties in the bottom quartile, was 1.23 (95% confidence interval (CI): 1.10, 1.37). Veterans living in the top quartile of counties with Medicaid eligibility were 1.17 (95% CI: 1.05, 1.37) times more likely to test positive for COVID-19 compared to Veterans living in the bottom quartile. Additionally, the relative risk of testing positive for COVID-19 among all Veterans living in the third quartile of crowded housing was 1.10 (95% CI: 1.04, 1.17) compared to the first quartile of persons in crowded housing. The association between county-level SDH and COVID-19 cases also varied in race-stratified models. The relative risk for testing positive for COVID-19 among Black Veterans living in counties in the top versus bottom quartile of percentage of persons who are non-White was 1.16 (95% CI: 1.01, 1.33); however, among White Veterans the RR was attenuated (1.08 (95% CI: 0.95, 1.10)). Among Black Veterans living in counties in the top versus bottom quartile of percentage of households with multigenerational housing, the risk of testing positive for COVID-19 was 1.14 (95% CI: 1.04, 1.25), yet among White Veterans the RR was 1.01 (95% CI: 0.93, 1.10). Among Other Veterans, living in a county in the top versus bottom quartile of percentage of residents 25 years or older without 4+ years of college education was associated with a 31% (95% CI: 1.09–1.59) higher risk of testing positive for COVID-19 versus the lowest quartile. Comparing the top versus the bottom quartile, little to no differences were seen among the percentage of persons in deep poverty, the percentage of persons without a computer or broadband, and the percentage of non-US-born residents.

## 4. Discussion

Our results demonstrate that Veterans living in areas with lower education levels, higher Medicaid eligibility, crowded housing, non-White residents, and multigenerational housing have higher risks of COVID-19 infection, a trend which has been noted in other evaluations [1,2,3,4]. Notably our assessment revealed important associations for our Veterans, such as percentage of residents who are non-White, living in multigenerational housing, and percentage of residents without a college degree varied in race-stratified models, strengthening for Black and Other Veterans, compared to White Veterans which provides important insights for our targeted interventions.

Our findings are consistent with previous research examining the associations between area-level socioeconomic-based measures and COVID-19 disparities in nationally representative samples, integrated healthcare systems, and data sources [18,29,30,31,32]. In an analysis of incidence and mortality data in the first 6 months of the pandemic, researchers examined disparities associated with county-level economic, housing, transit, population health, and health care characteristics. Main findings included significant associations between higher COVID-19 case and death counts and higher percentages of multi-unit households (Incidence Rate Ratio = IRR  =  1.02, 95% CI: 1.01–1.04), as well as higher percentages of individuals with limited English proficiency (IRR  =  1.09, 95% CI: 1.04–1.14) [30]. Moreover, in a cross-sectional analysis of U.S. census and combined statistical areas (CSAs) data, neighborhood race/ethnicity and poverty with COVID-19 infections and related deaths in urban US counties, researchers found excess burden of both infections and deaths was experienced by poorer and more diverse areas [29]. Similar findings were also found in an analysis of neighborhood-level measures of immigration, race, housing, and socio-economic characteristics with disparities in COVID-19 across Ontario, Canada [32].

In the present assessment, we simultaneously leverage community information with individual-level health data to provide additional insights beyond individual-level health data alone. Moreover, this evaluation provides additional support for making disaggregated SDH data more accessible for exploring causal mechanisms, including other social drivers of health. An additional strength of our work is that our findings demonstrate the association between distinct county-level SDH and COVID-19 cases, which was possible due to the large cohort size from a nationwide database from the largest integrated healthcare system in the United States. Moreover, our assessment was designed to provide a more precise evaluation of COVID-19 risk factors to direct targeted enhancement for patient care, which was also achieved by reducing confounding factors such as chronic health conditions. Chronic conditions are more common in our population and, thus, may attenuate the effects of individual-level socioeconomic and VA facility-level characteristics [28,33,34,35].

### Limitations

Our assessment has some limitations. First, our findings may have limited generalizability given that our evaluation is focused on understanding and improving the care of our unique Veteran population who are on average more likely to be male, older, and have multiple comorbidities compared to the general U.S. population. Second, the association between COVID-19 infection risk and Veterans’ county-level SDH may be stronger than the estimated results presented here, owing to the fact that some covariates adjusted for in this analysis may likely be mediators in the pathway, which would attenuate risk. Third, Veterans’ home addresses may not fully capture where Veterans spend most of their time, which may result in exposure misclassification. However, we anticipate misclassification would be attenuated by county-level aggregation. Fourth, while the SDH variables we examined in this retrospective analysis may be correlated (Appendix A), our large, diverse, and nationally representative sample of Veterans provides a robust and important window into disease risk. Fifth, while it is possible that the adjustment of some of these characteristics is attenuating the association between other factors due to the inter-related nature of these social determinants of health, these county-level social determinants of health are commonly examined together and noted as comprehensively covering socioeconomic and social drivers of various health outcomes including COVID-19 [1,3,4,5,6,7].

## 5. Conclusions

In this evaluation of Veterans enrolled at VA, we identified that county-level SDH factors influence COVID-19 infection risk, informing our understanding of how to improve care strategies, targeted interventions, policy, and resource allocation for Veterans. Understanding and eliminating individual and geographic disparities in COVID-19 has been identified as a national priority by the federal government and the recently established congressional COVID-19 Racial and Ethnic Disparities Task Force Act of 2020 (H.R.6763) roadmaps. Our findings may support county- and state-level policy makers in their response to COVID-19 by highlighting how area-level social determinants of health contribute to specific vulnerable populations’ overall burden of COVID-19. Augmenting individual social determinants of health data with detailed geographic social determinants of health data, therefore, provides unique opportunities to identify modifiable mechanisms by which area-level factors produce COVID-19 disparities, inform existing models for understanding COVID-19 disparities, and shape care policy.

## Figures and Tables

**Table 1 ijerph-18-13140-t001:** Individual and county-level demographic and social determinants of health characteristics among U.S. Veterans enrolled in active care in the Veterans Health Administration (VHA), 8 February–28 December 2020.

*n* (%)	All (*n* = 778,599)	White (*n* = 526,480)	Black *n* = 186,373	Other (*n* = 65,746)
**SARS-CoV-2 Test Result**				
Negative	700,907 (90.0)	476,642 (90.5)	165,708 (88.9)	58,557 (89.1)
Positive	77,692 (10.0)	49,838 (9.5)	20,665 (11.1)	7189 (10.9)
**Sex**				
Male	691,365 (88.8)	477,006 (90.6)	157,087 (84.3)	57,272 (87.1)
Female	87,234 (11.2)	49,474 (9.4)	29,286 (15.7)	8474 (12.9)
**Age**				
18–34	52,308 (6.7)	34,901 (6.6)	10,309 (5.5)	7098 (10.8)
35–44	78,661 (10.1)	51,726 (9.8)	17,560 (9.4)	9375 (14.3)
45–54	97,506 (12.5)	59,319 (11.3)	28,597 (15.3)	9590 (14.6)
55–64	164,301 (21.1)	96,631 (18.4)	54,890 (29.5)	12,780 (19.4)
65–74	248,054 (31.9)	177,486 (33.7)	53,397 (28.7)	17,171 (26.1)
75+	137,769 (17.7)	106,417 (20.2)	21,620 (11.6)	9732 (14.8)
**Race**				
White	526,480 (67.6)	--	--	--
Black/African American	186,373 (23.9)	--	--	--
Asian	9665 (1.2)	--	--	9665 (14.7)
American Indian/Alaska Native	7485 (1.0)	--	--	7485 (11.4)
Pacific Islander/Native Hawaiian	6874 (0.9)	--	--	6874 (10.5)
Unknown/Missing	41,722 (5.4)	--	--	41,722 (63.5)
**Ethnicity**				
Hispanic or Latino	57,801 (7.4)	40,944 (7.8)	3329 (1.8)	13,528 (20.6)
Not Hispanic or Latino	703,052 (90.3)	480,900 (91.3)	181,434 (97.3)	40,718 (61.9)
Unknown	17,746 (2.3)	4636 (0.9)	1610 (0.9)	11,500 (17.5)
**Marital Status**				
Single	130,442 (16.8)	76,932 (14.6)	40,493 (21.7)	13,017 (19.8)
Married	365,781 (47.0)	264,479 (50.2)	69,966 (37.5)	31,336 (47.7)
Divorced/Separated/Widowed	273,214 (35.1)	180,150 (34.2)	74,048 (39.7)	19,016 (28.9)
**Urban/Rural**				
Urban	573,072 (73.6)	355,834 (67.6)	164,582 (88.3)	52,656 (80.1)
Rural	205,527 (26.4)	170,646 (32.4)	21,791 (11.7)	13,090 (19.9)
**Region**				
Continental	114,974 (14.8)	75,113 (14.3)	30,589 (16.4)	9272 (14.1)
Midwest	160,422 (20.6)	121,541 (23.1)	29,694 (15.9)	9187 (14.0)
Northeast	184,049 (23.6)	119,302 (22.7)	54,442 (29.2)	10,305 (15.7)
Pacific	155,350 (20.0)	105,629 (20.1)	22,431 (12.0)	27,290 (41.5)
Southeast	163,804 (21.0)	104,895 (19.9)	49,217 (26.4)	9692 (14.7)
**Priority Group ^a^**				
No Service Disability	89,103 (11.4)	62,920 (12.0)	18,647 (10.0)	7536 (11.5)
Low Income	154,047 (19.8)	105,635 (20.1)	37,995 (20.4)	10,417 (15.8)
Low/Moderate Disability	151,646 (19.5)	108,431 (20.6)	30,684 (16.5)	12,531 (19.1)
High Disability	383,803 (49.3)	249,494 (47.4)	99,047 (53.1)	35,262 (53.6)
**Median (P25–P75)**	**All** **(*n* = 778,599)**	**White** **(*n* = 526,480)**	**Black** ***n* = 186,373**	**Other** **(*n* = 65,746)**
Percentage without High School Diploma, Ages 25+	11.8 (9.3–14.7)	11.4 (9.0–14.4)	12.4 (9.7–15.5)	12.6 (9.7–16.3)
Percentage without 4+ Years College, Ages 25+	69.9 (65.1–77.8)	70.6 (65.6–78.9)	68.8 (62.8–75.7)	69.1 (64.0–77.2)
Percentage Food Stamps/SNAP Recipients	14.6 (10.8–18.3)	14.1 (10.3–17.2)	15.8 (12.3–20.4)	13.7 (9.7–17.9)
Percentage without Health Insurance, Under Age 65	9.7 (6.6–12.8)	9.4 (6.3–12.6)	10.7 (7.9–13.0)	9.6 (7.1–12.7)
Percentage Eligible for Medicaid, All Ages, 2012	22.9 (18.0–27.3)	22.5 (17.7–26.2)	23.6 (19.4–28.7)	24.4 (18.9–28.9)
Percentage in Crowded Housing	2.36 (1.6–3.8)	2.22 (1.5–3.5)	2.46 (1.71–3.81)	3.6 (2.01–6.54)
Percentage 65+ living alone	10.8 (9.0–12.3)	11.0 (9.1–12.6)	10.3 (8.8–11.6)	9.6 (8.8–11.5)
Percentage of Households without a computer	10.7 (8.4–14.1)	10.8 (8.5–14.2)	10.7 (8.5–14.2)	9.6 (7.5–12.3)
Percentage of households without broadband	19.1 (15.6–23.5)	19.2 (15.6–23.6)	19.5 (16.1–23.7)	17.9 (14.4–21.8)
Percentage Non-US-born residents	8.6 (4.5–16.7)	7.7 (3.8–14.9)	9.8 (5.6–21.1)	13.3 (6.6–23.3)
Percentage Non-White	36.2 (19.9–54.9)	28.6 (15.2–47.6)	50.7 (36.24–63.36)	49.7 (28.2–63.9)
Median Household Income (thousands)	54.5 (47.6–62.4)	54.5 (47.6–62.3)	54.4 (46.7–61.6)	57.6 (50.3–68.9)
Unemployment Rate Ages 16+, 2017	4.1 (3.6–4.7)	4.0 (3.5–4.7)	4.3 (3.7–4.9)	4.0 (3.5–4.7)
Percentage of households where grandparent have children under 18	5.9 (4.5–7.3)	5.52 (4.3–7.2)	6.4 (4.97–7.51)	6.75 (5.0–8.5)
Percentage Persons 65+ in Deep Poverty	2.8 (2.3–3.4)	2.7 (2.2–3.3)	3.1 (2.5–3.6)	3.0 (2.4–3.4)
Percentage of Persons in Deep Poverty	6.7 (5.2–7.7)	6.5 (5.0–7.5)	7.2 (6.0–8.7)	6.8 (5.3–7.5)

^a^ Priority group refers to a priority-based enrollment system enacted in 1996 to ensure the Veterans are enrolled based on ranked eligibility status: service-connected disability rating, income, recent military service, and other factors. Abbreviations: P25 25th percentile; P75, 75th percentile.

**Table 2 ijerph-18-13140-t002:** Adjusted risk ratios (95 CI) for receiving a positive COVID-19 test result among Veterans enrolled in active care at the Veterans Health Administration (VHA) who obtained a COVID-19 test, 8 February–28 December 2020 ^a^.

		All (*n* = 778,599)		White (*n* = 526,480)		Black (*n* = 186,373)		Other (*n* = 65,746)	
Percentage of Persons in Deep Poverty	3.5–11.2	REF	--	REF	--	REF	--	REF	--
	>11.2–14.7	0.96	[0.89, 1.02]	0.97	[0.90, 1.04]	0.88	[0.79, 0.98]	0.96	[0.84, 1.10]
	>14.7–17.0	0.94	[0.86, 1.03]	0.95	[0.87, 1.05]	0.89	[0.78, 1.01]	0.89	[0.76, 1.04]
	>17.0–48.6	0.94	[0.83, 1.05]	0.96	[0.84, 1.09]	0.87	[0.73, 1.03]	0.87	[0.72, 1.04]
Percentage without 4+ Years College, Ages 25+	21.9–65.1	REF	--	REF	--	REF	--	REF	--
	>65.1–69.9	1.14	[1.08, 1.22]	1.19	[1.12, 1.26]	1.06	[0.98, 1.14]	1.08	[0.95, 1.23]
	>69.9–77.8	1.12	[1.04, 1.21]	1.16	[1.07, 1.26]	1.02	[0.93, 1.12]	1.12	[0.98, 1.28]
	>77.8–95.1	1.23	[1.10, 1.37]	1.24	[1.11, 1.38]	1.14	[0.99, 1.31]	1.31	[1.09, 1.59]
Percentage Food Stamps/SNAP Recipients	0.4–10.8	REF	--	REF	--	REF	--	REF	--
	>10.8–14.6	1.01	[0.94, 1.09]	1.00	[0.92, 1.09]	1.05	[0.96, 1.15]	1.06	[0.96, 1.18]
	>14.6–18.3	1.07	[0.99, 1.16]	1.05	[0.96, 1.15]	1.10	[0.98, 1.24]	1.08	[0.94, 1.24]
	>18.3–57.3	1.06	[0.95, 1.18]	1.06	[0.94, 1.21]	0.97	[0.83, 1.13]	1.18	[0.98, 1.43]
Percentage without Health Insurance, Under Age 65	2.1–6.5	REF	--	REF	--	REF	--	REF	--
	>6.5–9.7	0.97	[0.92, 1.02]	0.95	[0.86, 1.04]	0.99	[0.92, 1.07]	0.86	[0.76, 0.99]
	>9.7–12.8	0.92	[0.86, 0.98]	0.84	[0.68, 1.04]	1.04	[0.90, 1.21]	0.79	[0.69, 0.92]
	>12.8–31.1	0.94	[0.84, 1.06]	0.71	[0.46, 1.10]	1.21	[1.00, 1.47]	0.77	[0.65, 0.91]
Percentage Eligible for Medicaid	0.6–18.0	REF	--	REF	--	REF	--	REF	--
	>18.0–22.9	1.04	[0.97, 1.11]	1.01	[0.94, 1.01]	1.06	[0.99, 1.14]	1.12	[1.00, 1.24]
	>22.9–27.3	1.08	[0.99, 1.22]	1.07	[0.97, 1.17]	1.10	[0.99, 1.22]	1.11	[0.99, 1.25]
	>27.3–62.0	1.17	[1.05, 1.37]	1.16	[1.02, 1.32]	1.20	[1.05, 1.37]	1.21	[1.03, 1.44]
Percentage in Crowded Housing	0.0–1.6	REF	--	REF	--	REF	--	REF	--
	>1.6–2.4	1.01	[0.96, 1.07]	0.99	[0.94, 1.06]	1.04	[0.98, 1.10]	1.01	[0.91, 1.13]
	>2.4–3.8	1.10	[1.04, 1.17]	1.11	[1.03, 1.19]	1.07	[1.00, 1.15]	1.08	[0.95, 1.21]
	>3.8–34.9	1.07	[0.99, 1.17]	1.10	[0.99, 1.21]	0.99	[0.88, 1.11]	1.12	[0.93, 1.34]
Percentage 65+ living alone	2.8–9.0	REF	--	REF	--	REF	--	REF	--
	>9.0–10.8	0.97	[0.92, 1.03]	2.13	[0.79, 5.73]	0.93	[0.84, 1.03]	0.95	[0.88, 1.03]
	>10.8–12.3	0.98	[0.91, 1.05]	2.21	[0.76, 6.44]	0.97	[0.81, 1.15]	1.01	[0.92, 1.10]
	>12.3–31.8	0.94	[0.87, 1.02]	2.18	[0.76, 6.25]	0.93	[0.78, 1.09]	0.90	[0.82, 0.99]
Percentage Without Computer	1.4–8.4	REF	--	REF	--	REF	--	REF	--
	>8.4–10.7	1.04	[0.97, 1.13]	1.08	[0.99, 1.18]	0.98	[0.93, 1.05]	1.07	[0.94, 1.22]
	>10.7–14.1	1.06	[0.96, 1.18]	1.09	[0.96, 1.23]	1.07	[0.97, 1.17]	1.09	[0.90, 1.32]
	>14.1–61.7	1.05	[0.93, 1.19]	1.08	[0.94, 1.24]	1.06	[0.92, 1.21]	1.09	[0.86, 1.37]
Percentage Without Broadband Internet	6.0–15.6	REF	--	REF	--	REF	--	REF	--
	>15.6–19.1	1.04	[0.97, 1.12]	1.02	[0.94, 1.10]	1.06	[0.97, 1.15]	1.05	[0.91, 1.21]
	>19.1–23.5	1.02	[0.92, 1.13]	0.99	[0.88, 1.11]	1.07	[0.94, 1.21]	0.97	[0.80, 1.17]
	>23.5–74.3	1.03	[0.92, 1.16]	1.01	[0.88, 1.14]	1.06	[0.93, 1.21]	0.91	[0.72, 1.16]
Percentage Non-US-born residents	0.0–4.5	REF	--	REF	--	REF	--	REF	--
	>4.5–8.6	1.02	[0.96, 1.07]	1.01	[0.95, 1.07]	1.06	[0.98, 1.15]	0.94	[0.84, 1.06]
	>8.6–16.7	1.01	[0.94, 1.10]	0.99	[0.91, 1.08]	1.07	[0.96, 1.19]	0.99	[0.86, 1.14]
	>16.7–53.3	1.15	[1.03, 1.29]	1.12	[1.00, 1.25]	1.14	[0.98, 1.32]	1.19	[0.99, 1.43]
Percentage Non-White	1.7–19.5	REF	--	REF	--	REF	--	REF	--
	>19.5–36.2	1.04	[0.98, 1.11]	1.07	[1.00, 1.14]	1.05	[0.94, 1.17]	1.03	[0.93, 1.15]
	>36.2–55.0	1.01	[0.92, 1.10]	1.01	[0.92, 1.10]	1.09	[0.97, 1.22]	0.98	[0.83, 1.16]
	>55.0–97.2	1.11	[0.99, 1.24]	1.08	[0.95, 1.22]	1.16	[1.01, 1.33]	1.13	[0.92, 1.38]
Median Household Income (thousands)	22.0–47.6	REF	--	REF	--	REF	--	REF	--
	>47.6–54.5	0.97	[0.91, 1.03]	0.98	[0.91, 1.05]	0.94	[0.87, 1.01]	0.94	[0.83, 1.07]
	>54.5–62.4	1.05	[0.98, 1.13]	1.04	[0.95, 1.14]	1.06	[0.95, 1.18]	0.99	[0.85, 1.16]
	>62.4–134.6	1.03	[0.91, 1.17]	1.01	[0.88, 1.17]	1.03	[0.90, 1.18]	0.95	[0.78, 1.15]
Unemployment Rate Ages 16+, 2017	1.3–3.6	REF	--	REF	--	REF	--	REF	--
	>3.6–4.1	1.02	[0.97, 1.07]	0.81	[0.63, 1.03]	0.99	[0.91, 1.08]	1.03	[0.97, 1.10]
	>4.1–4.7	0.98	[0.92, 1.04]	0.87	[0.67, 1.13]	0.93	[0.85, 1.02]	0.98	[0.90, 1.07]
	>4.7–20.5	0.93	[0.86, 1.01]	0.77	[0.56, 1.06]	0.86	[0.78, 0.94]	0.96	[0.85, 1.09]
Percentage of households where grandparents have children under 18	0.0–4.5	REF	--	REF	--	REF	--	REF	--
	>4.5–5.9	1.06	[1.00, 1.12]	1.07	[1.00, 1.13]	1.10	[1.03, 1.19]	0.95	[0.85, 1.06]
	>5.9–7.3	1.05	[0.98, 1.12]	1.05	[0.97, 1.13]	1.17	[1.08, 1.28]	0.93	[0.81, 1.06]
	>7.3–32.4	1.01	[0.93, 1.08]	1.01	[0.93, 1.10]	1.14	[1.04, 1.25]	0.89	[0.76, 1.05]

^a^ Adjusted for individual demographics (race/ethnicity, facility characteristics, Gini coefficient, and rural/urban/highly rural).

## Data Availability

Due to US Department of Veterans Affairs (VA) regulations and our ethics agreements, the analytic data sets used for this assessment are not permitted to leave the VA firewall without a Data Use Agreement. This limitation is consistent with other studies based on VA data. However, VA data are made freely available to researchers with an approved VA study protocol. For more information, please visit https://www.virec.research.va.gov (accessed on 1 February 2021) or contact the VA Information Resource Center at VIReC@va.gov.

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
