# Peer review of "Differences in COVID-19 Risk by Race and County-Level Social Determinants of Health among Veterans"

_ijerph, 2021, doi:10.3390/ijerph182413140_

Round 1

Reviewer 1 Report

The paper is interesting, offering new evidence on inequalities in the risk of positivity to Covid-19. In my opinion the paper offers a worthy contribution to the literature on inequalities in the US society, which could have policy implications, and I appreciated it. Nevertheless, I have some doubts and suggestion to share with the Authors.

I acknowledge that this is a brief report, but large space of the paper is devoted to the presentation of the data and results (sections 1-3), whereas I found the discussion section too brief and general. I would ask the Authors to deepen and enlarge the discussion section, in order to better explain the differences they found. In this version is too vague and general (e.g., the comments on multigenerational households or crowded housing are too synthetic and not completely correct in my opinion). The Authors should better resume and present to the reader their main findings.

As for the model specification, if I understand correctly, also individual demographic characteristics are accounted for in the models (see, e.g., the note a to table 2). I think that it would be interesting to see some of this results, to appreciate the different contribution of both individual and contextual characteristics to the different Covid-19 risk.

Moreover (and this is my major concern), I wonder if there are too many contextual characteristics in the model specification, which are somewhat correlated the one with the other. Could this explain the lack of significance of some aspects? Have the Authors tried different model specifications? Have they evaluated potential correlation between macro-level factors?

The models are also controlled for some contextual aspects not reported in the Table 2 (e.g., unemployment rate, percentage of elderly living alone, of people without health insurance, and so on). Why these results are not reported? Indeed, in the tables there are some aspects which are never significant, so I cannot really understand the logic of the presentation of these results.

Finally, I would ask the Authors to further discuss the policy implications introduced in the concluding section. I found it somewhat vague and underdeveloped.

Please check the first sentence of “Limitations”.

Author Response

Response to Reviewer 1 Comments

Point 1: The paper is interesting, offering new evidence on inequalities in the risk of positivity to Covid-19. In my opinion the paper offers a worthy contribution to the literature on inequalities in the US society, which could have policy implications, and I appreciated it. Nevertheless, I have some doubts and suggestion to share with the Authors.

Response 1: Thank you for your detailed and thoughtful review of our research. We appreciate you taking the time to review our paper. We have been able to address all your comments, as noted below. 

Point 2: I acknowledge that this is a brief report, but large space of the paper is devoted to the presentation of the data and results (sections 1-3), whereas I found the discussion section too brief and general. I would ask the Authors to deepen and enlarge the discussion section, in order to better explain the differences they found. In this version is too vague and general (e.g., the comments on multigenerational households or crowded housing are too synthetic and not completely correct in my opinion). The Authors should better resume and present to the reader their main findings.

Response 2: We have revised the discussion, accordingly, expanding on our main findings. The following sentences have been included in the discussion section. “Our findings are consistent with previous research examining the associations between area-level socioeconomic based measures and COVID-19 disparities in nationally-representative samples, integrated healthcare systems, and data sources.11,14-17 In an analysis of incidence and mortality data in the first 6 months of the pandemic, prior research examined disparities associated with county-level economic, housing, transit, population health and health care characteristics. The primary findings included significant associations between higher COVID-19 case and death counts and higher percentages of multi-unit households (Incidence Rate Ratio=IRR = 1.02, 95% CI: 1.01–1.04), higher percentages of individuals with limited English proficiency (IRR = 1.09, 95% CI: 1.04–1.14).15 Moreover, in a cross-sectional analysis of U.S. census and combined statistical areas (CSAs) data, neighbourhood race/ethnicity and poverty with COVID-19 infections and related deaths in urban US counties, researchers found excess burden of both infections and deaths was experienced by poorer and more diverse areas.14 Similar findings were also found in an analysis of neighbourhood-level measures of immigration, race, housing and socio-economic characteristics with disparities in COVID-19 across Ontario, Canada.17

Point 3: As for the model specification, if I understand correctly, also individual demographic characteristics are accounted for in the models (see, e.g., the note a to table 2). I think that it would be interesting to see some of this results, to appreciate the different contribution of both individual and contextual characteristics to the different Covid-19 risk.

Response 3: Our group, previously published a manuscript focusing on the individual demographic characteristics (published in Public Health Reports) associated with COVID-19 risk among VA population. This is another reason why, in this paper, we wanted to focus on the contribution of the contextual social determinants of health associated with COVID-19 risk among VA population. We have highlighted some of these findings in the results section. “In addition to examining county-level contextual factors, fully adjusted models adjusted for individual demographics and facility characteristics, Gini coefficient, percentage aged 65+ living alone, rural/urban/highly rural, unemployment rate (2017), without health insurance (data not shown). Consistent with our previous findings, in fully adjusted models we found that female, Black, urban, low-income, and disabled Veterans were more likely to test positive for COVID-19.25 These disparities are also consistent with other studies examining differences in COVID-19 testing and test positivity within the VA population.2,23

Point 4: Moreover (and this is my major concern), I wonder if there are too many contextual characteristics in the model specification, which are somewhat correlated the one with the other. Could this explain the lack of significance of some aspects? Have the Authors tried different model specifications? Have they evaluated potential correlation between macro-level factors?

Response 4: Thank you for noting this. Yes, we have previously evaluated the potential correlations between the county-level social determinants of health in our study. We have now included Supplementary Table 2 presenting the correlation between the county-level social determinants of health in our analysis. For multiple reasons, we did not examine additional model specifications. First, we didn’t see any issues with the levels of correlations in between our variables. This is likely because high correlation tends to cause model failure in small sample sizes due to one value predicting another, but we have a large sample size (N=778,599).  Relatedly, when we examined the correlations for the variables in our models, most of the coefficients were not high (Supplementary Table 2). Second, we included these specific variables based completely on an a priori assessment of the literature and conceptualizations of our model specifications. While it is possible that the adjustment of some of these characteristics is attenuating the association between other factors due to the inter-related nature of these social determinants of health, these county-level social determinants of health are commonly examined together and noted as a comprehensively covering socioeconomic and structural measures of social drivers and determinants of various health outcomes. Namely, these measures have been used in nationally representative samples of COVID outcomes (e.g., Chin 2020, Krieger 2020, and Basset 2020). Moreover, in the methods section, we had previously including the following sentence: “In brief, we included demographic characteristics from the VA’s EHR database and used the Veteran’s home zip code to geographically link publicly available area-based SDH as it has been previously identified being critical for COVID-19 health equity in previous literature.1,20,27-30” We have added this reasoning to the limitations section to support our interpretations of our findings: “Fifth, while the SDH variables we examined may be correlated (Supplementary Table 2), this did not present an issue in our study due to our large, diverse, and nationally representative sample of Veterans. Moreover, while it is possible that the adjustment of some of these characteristics is attenuating the association between other factors due to the inter-related nature of these social determinants of health, these county-level social determinants of health are commonly examined together and noted as a comprehensively covering socioeconomic and social drivers of various health outcomes including COVID-19.1,6,7,35

Point 5: The models are also controlled for some contextual aspects not reported in the Table 2 (e.g., unemployment rate, percentage of elderly living alone, of people without health insurance, and so on). Why these results are not reported? Indeed, in the tables there are some aspects which are never significant, so I cannot really understand the logic of the presentation of these results.

Response 5: Thank you for this suggestion. The models are adjusted for individual demographics and facility characteristics, Gini coefficient for income inequality, Percentage 65+ living alone, rural/urban/highly rural, Unemployment Rate Ages 16+, 2017, and Percentage without Health Insurance, Under Age 65. We have added the results for Percentage 65+ living alone, Unemployment Rate Ages 16+, 2017, and Percentage without Health Insurance, and the population under age 65. The results for facility characteristics, rural/urban/highly rural, and Gini coefficient for income inequality are presented in our previously published paper in Public Health Reports. One rationale for including items as confounders and not presenting in regression output is that they are relevant for confounder control but were not the a priori variables of interest. For example, results for urban/rural/highly rural, individual demographics, and facility characteristics are already included in our focused analysis published in Public Health Reports. For the GINI coefficient, we believe that this requires specific analysis. For example, first author HS Abdel Magid has recently co-authored a paper on Gini and COVID in JAMA Network Open, (see Tan 2021). Similarly, as unemployment rate is historical, we didn’t want readers to interpret it incorrectly as being current unemployment rate. Nevertheless, we have included the results for Percentage of people 65+ and living alone, Unemployment Rate Ages 16+ [specify that 2017 is the year of the unemployment rate], 2017, and Percentage without Health Insurance, and population under age 65 in Table 2 as per the reviewer’s suggestions.

Point 6: Finally, I would ask the Authors to further discuss the policy implications introduced in the concluding section. I found it somewhat vague and underdeveloped.

Response 6: We have added the following sentences to the conclusion section to highlight some of the policy implications of our findings: “Understanding and eliminating individual and geographic disparities in COVID-19 has been identified as a national priority by the federal government and the recently established congressional COVID-19 Racial and Ethnic Disparities Task Force Act of 2020 (H.R.6763) roadmaps. Our findings may support county- and state-level policy makers in their response to COVID-19 by highlighting how area-level social determinants of health contribute to vulnerable populations overall burden of COVID-19. Augmenting individual social determinants of health data with detailed geographic social determinants of health data, therefore, provides unique opportunities to identify modifiable mechanisms by which area-level factors produce COVID-19 disparities, inform existing models for understanding COVID-19 disparities, and shape policy.”

Point 7: Please check the first sentence of “Limitations”.

Response 7: We have clarified the first sentence of the limitations. This now reads as “Our study has some limitations. First, our findings may have limited generalizability given that our evaluation is focused on evaluating our unique Veteran population, who are on average are male, older, and have more comorbidities than the general US population, which limits generalizability.15

Reviewer 2 Report

This manuscript describes disparities in COVID-19 infection by area level SDH impacting U.S. Veterans. The authors evaluate the retrospective analysis of COVID-19 data and found an exposure response relationship with increase in risk for each county level SDH such as county without a college degree and living in crowded housing etc. This is an organized manuscript that would be recommended for acceptance. Suggestions to improve this manuscript are listed below:

  1. The introduction section the authors need to address any other references if conducted for a similar type of study if available for COVID
  2. Standard error can be included in the table
  3. Grammatical errors need to be checked throughout the manuscript eg: line 120
  4. For supplementary table 1 , including a link for the source would be ideal
  5. Please maintain the reference style format to be consistent for all the references.
    1. Some references do not have full journal name
    2. Some references do not have volume and page numbers

Author Response

Response to Reviewer 2 Comments

Point 1: This manuscript describes disparities in COVID-19 infection by area level SDH impacting U.S. Veterans. The authors evaluate the retrospective analysis of COVID-19 data and found an exposure response relationship with increase in risk for each county level SDH such as county without a college degree and living in crowded housing etc. This is an organized manuscript that would be recommended for acceptance. Suggestions to improve this manuscript are listed below:

Response 1: Thank you for your detailed and thoughtful review of our research. We appreciate you taking the time to review our paper. We have been able to address all of your comments, as noted below. 

Point 2: The introduction section the authors need to address any other references if conducted for a similar type of study if available for COVID

Response 2: We have significantly expanded our references list, simultaneously highlighting previous literature in this space and the gaps we fill with our study. Our references list now includes 36 of the most pertinent references in this area of research. We have added the following sections to the introduction: “Disparities in COVID-19 infection and mortality vary across the US.1-3 These disparities, particularly among racial and ethnic minorities, may be driven by area-level social determinants of health (SDH) and structural resources.4-18 For example, higher income inequality at state19 and county20 levels have been linked to increased COVID-19 burden. Income inequality may exacerbate opportunities for infection as the most disadvantaged individuals need to stay in the labor force to afford to live in a region that also includes much wealthier residents.21 Moreover, lower income individuals are more likely to reside in crowded housing and have public-facing jobs such as service, child and elder care, and cleaning/janitorial services, which can increase the risk of exposure.22 Although these studies, have provided evidence necessary to understand the area-level SDH associated with COVID-19 disparities, the examination of these relationships in integrated healthcare systems has been limited. Indeed, to our knowledge, the association between county-level SDH and COVID-19 among Veterans has not been previously examined. Health systems are the focal point of the COVID-19 pandemic, and vital to understanding the extent of the pandemic and identifying groups at highest risk. The electronic health record database of the Department of Veterans Affairs (VA) offers the single largest national data resource available with the necessary information on system-wide testing and detailed medical histories to examine racial and ethnic disparities in the US. Recent research suggests that some racial and ethnic minorities as well as socioeconomically disadvantaged groups in the VA are bearing a disproportionate burden of COVID-19.23-25

Point 3: Standard error can be included in the table

Response 3: In table 2, we provide the 95% CI, which is the standard for reporting results of a regression. Considering, our limited word and space availability for a brief report, we prioritized reporting the field standard of the 95% CI over reporting standard errors.

Point 4: Grammatical errors need to be checked throughout the manuscript eg: line 120

Response 4: Thank you for the reminder. We have thoroughly checked grammatical errors throughout our manuscript, including in line 120. Line 120 has now been corrected, and reads “Moreover, our assessment was designed to provide a more precise evaluation to direct targeted enhancement for our patients which was also achieved by reducing confounding factors from chronic health conditions. Chronic conditions are more common in our population and thus, may attenuate the effects of individual-level socioeconomic and VA facility-level characteristics.”

Point 5: For supplementary table 1, including a link for the source would be ideal

Response 5: We have added the following sentence as a footnote to supplementary table 1:  “Data were retrieved from the American Community Survey, U.S. Census Bureau, U.S. Bureau of Labor Statistics, and Diversity Data for Kids as previously described in Chin et al 2020, and available at https://bmjopen.bmj.com/content/10/9/e039886”

Point 6: Please maintain the reference style format to be consistent for all the references. Some references do not have full journal name. Some references do not have volume and page numbers.

Response 6: We have corrected all references, ensuring consistency across our references list.

This manuscript is a resubmission of an earlier submission. The following is a list of the peer review reports and author responses from that submission.